# Simultaneous Measurement of Flow Velocity and Electrical Conductivity of a Liquid Metal Using an Eddy Current Flow Meter in Combination with a Look-Up-Table Method

**DOI:** 10.3390/s23229018

**Published:** 2023-11-07

**Authors:** Nico Krauter, Frank Stefani

**Affiliations:** Helmholtz-Zentrum Dresden-Rossendorf, Bautzner Landstraße 400, 01328 Dresden, Germany; n.krauter@hzdr.de

**Keywords:** eddy current, inductive flow measurement, look-up-table, liquid metal

## Abstract

The Eddy Current Flow Meter (ECFM) is a commonly employed inductive sensor for assessing the local flow rate or flow velocity of liquid metals with temperatures up to 700 ${}^{\circ }$C. One limitation of the ECFM lies in its dependency on the magnetic Reynolds number for measured voltage signals. These signals are influenced not only by the flow velocity but also by the electrical conductivity of the liquid metal. In scenarios where temperature fluctuations are significant, leading to corresponding variations in electrical conductivity, it becomes imperative to calibrate the ECFM while concurrently monitoring temperature to discern the respective impacts of flow velocity and electrical conductivity on the acquired signals. This paper introduces a novel approach that enables the concurrent measurement of electrical conductivity and flow velocity, even in the absence of precise knowledge of the liquid metal’s conductivity or temperature. This method employs a Look-Up-Table methodology. The feasibility of this measurement technique is substantiated through numerical simulations and further validated through experiments conducted on the liquid metal alloy GaInSn at room temperature.

## 1. Introduction

The measurement of flow velocities in liquid metal is generally difficult due to their opacity and in many cases further aggravated by their high temperature and chemical reactivity. In some sense, however, these drawbacks are compensated by the high electrical conductivity of liquid metals which allows for the application of inductive flow measurement techniques. These include the Contactless Inductive Flow Tomography (CIFT) for the global reconstruction of entire two- or three-dimensional flow fields [1], and quite a few techniques for the measurement of local velocities, including the phase-shift sensor [2], the magnetic distortion probe [3], Lorentz force velocimetry [4,5,6,7,8], and Eddy Current Flow Meters (ECFM) [9,10,11,12,13,14,15,16,17,18]. The latter sensors are widely used in liquid metal cooled fast breeder reactors (LMFBR) as part of the reactor safety instrumentation to monitor the coolant flow velocity. Usually they consist of three solenoid coils: one primary coil that generates an alternating magnetic field and two passive secondary coils above and below the primary coil, which detect flow induced changes in the primary field [11] (see Figure 1). These changes are caused by flow induced eddy currents, which lead to an asymmetry of the eddy current distribution around the sensor and thus also to different induced voltages within the secondary coils. By measuring the difference between the magnitude and/or phase [19] of the secondary coil voltages, the flow velocity of the liquid metal can be inferred.

A significant drawback of the ECFM is that the output signal depends not only on the fluid velocity but also on the electrical conductivity, which in turn depends on the temperature of the liquid metal [20]. Typically, the coolant temperature of LMFBR is not constant, so that an extensive calibration of the ECFM is necessary for different temperatures. For small changes in the electrical conductivity of the liquid metal a temperature compensation can be performed [21], but this method only works for a certain excitation frequency. The Transient Eddy Current Flow Meter (TECFM) [22,23] overcomes the temperature-dependence problem by tracking the position of an impressed eddy current system which is moving with the same velocity as the conductive fluid. Yet, there are a couple of applications, in particular in LMFBR’s, in which a complementary determination of the temperature would also be desirable.

In this paper we present a new method for the simultaneous determination of flow velocity *v* and electrical conductivity σ of the liquid metal when using an ECFM. This technique relies on the creation of a so-called Look-Up-Table (LuT) which contains a large number of output voltages for different parameter combinations of *v*, σ and excitation frequencies *f* that were obtained in advance by numerical simulation of the sensor and its surroundings. By calculating the deviations between a given measurement result (with unknown *v* and σ) and each value in the LuT, a certain parameter combination of velocity and conductivity can be identified for which the deviation between measurement and simulation is minimal. Here, the method of least squares is used to represent this deviation in the form of a mean squared error (MSE). The parameter combination with the lowest MSE is most likely the unknown parameter combination, assuming that the numerical model depicts reality with sufficient accuracy. Up to three excitation frequencies are used in order to increase the accuracy of this method. For the level detection of liquid magnesium in the titanium production process the LuT method has already yielded promising results [24,25]. Here we will apply a similar technique to the ECFM, demonstrating how to simultaneously obtain *v* and σ by a detailed analysis of the numerical simulations. We will also present first results of measurements that were performed in the liquid metal alloy GaInSn.

## 2. Materials and Methods

### 2.1. Sensor and Measurement Setup

The ECFM sensor which is used for the experiments has a total height of 50 mm and an outer diameter of 10.6 mm. The secondary coils S1 and S2 each have a height of 16 mm and 1150 turns of enamelled copper wire with a diameter of 0.15 mm. The primary coil P1 has a height of 10 mm and 290 turns of enamelled copper wire with a diameter of 0.25 mm. The distance between two neighbouring coils is 2 mm. All coils are wound on a PVC coil holder which is screwed to a stainless steel threaded rod with a diameter of 3 mm (see Figure 1a,b). The primary coil is supplied with an alternating current with an amplitude of 280 mA and frequencies *f* of 500 Hz, 1 kHz, and 2 kHz. This ECFM is most sensitive to changes of the flow velocity in this frequency range. The coil voltages are measured with a Lock-In-Amplifier (LIA) which has the advantage that voltages that do not have a certain frequency, in this case the excitation frequency *f*, are filtered out and thus electromagnetic disturbances with a broad frequency spectrum have a strongly reduced impact on the measurement results. This enables highly accurate measurements of the coil voltages. The LIA also provides the amplitude and frequency of the excitation current, which is supplied to the primary coil via a power amplifier that ensures a constant current amplitude.

The experiments are performed in a liquid metal loop (see Figure 2) that uses the liquid metal alloy GaInSn, which has an electrical conductivity of 3.3 MS/m [26] and is liquid at room temperature. The chemical composition of the alloy is 67% gallium, 20.5% indium and 12.5% tin. The flow velocity of the liquid metal can be adjusted with a permanent-magnet pump that is connected to an electric motor. Velocities up to ±1.2 m/s can be achieved in the horizontal test section, which consists of a non-conductive PVC pipe with an inner diameter of 27 mm. A stainless steel sensor casing consisting of a tube, sealed on one side with a conical tip, an outer diameter of 13 mm and a wall thickness of 1 mm, is inserted into the GaInSn loop from the side, such that it is positioned on the axis of the pipe. The ECFM is inserted into the sensor casing and therefore has no direct contact with the liquid metal. The flow velocity of the loop $v_{\exp}$ is lower than the flow velocity around the ECFM $v_{\mathrm{ECFM}}$ since the sensor casing is effectively reducing the cross section of the test section. Therefore, the volume averaged flow velocity $v_{\mathrm{ECFM}}$ is 29.4% higher than $v_{\exp}$. This was confirmed with numerical simulations of the flow inside the GaInSn loop. In the following sections $v_{\exp}$ is used as the reference flow velocity, since this is the quantity that is supposed to be measured by the ECFM. The commercially available magnetic-inductive flow rate sensor ABB Copa-XL is used as a reference flow meter for the measurements.

The LIA measures the magnitude *r* and the phase shift φ of the coil voltage *V*. The interrelation between these values, using S1 as an example, is as follows: (1)$$V_{S1}=r_{S1}\cdot e^{j\varphi_{S1}}.$$

### 2.2. Calibration of the Numerical Simulation Model

Since a numerical simulation model is used to create the LuT, it is of utmost importance that the representation of the ECFM in the model is as close as possible to the real ECFM. Initially, a very simple 2D axisymmetric model of the sensor (see Figure 1c) is created. It contains the three coils of the ECFM in an infinite volume of air and the stainless steel rod at the centre of the sensor. The dashed red line in Figure 1c indicates the symmetry axis of the model. Using a 2D model has the advantage that the calculations can be performed much faster than with a 3D model, and since there are many thousands of calculations necessary to create the LuT, the model has to be as simple as possible. For comparable setups, the 2D model computes 30 times faster than a 3D model with the same geometry. Obviously, problems or setups that cannot be represented by axisymmetrical models, like bubble detection with the ECFM [27] for example, would require a 3D model. The numerical model is created with the data from the technical drawings and specifications of the ECFM. However, due to manufacturing tolerances, the induced voltages of the ECFM model might be slightly different to those of the real ECFM. Therefore, the model has to be calibrated by adjusting certain geometrical and electrical parameters to match those of the real sensor.

First, the inductance *L* for each of the three coils is measured and compared to the numerically obtained inductances. At this stage, the ECFM has not been inserted into the test section. Differences between model and real sensor inductances can be corrected by adjusting the number of wire turns *n* in the numerical model, until the simulated and measured inductances match. In Table 1 the number of turns according to the specifications $n_{\mathrm{spec}}$ and the calibrated number of turns $n_{\mathrm{cal}}$ are listed.

In the next step of calibration, the excitation current is supplied to the primary coil P1. The induced voltages $V_{S1}$ and $V_{S2}$ in the secondary coils are compared to the induced voltages of the numerical model. Since the inductances of the coils have already been corrected, any remaining voltage differences are most likely caused by a vertical displacement $\Delta d_{P1}$ of the secondary coils with respect to P1, which can be added to the model. The distances of the secondary coils to P1 are adjusted in the model, until the simulated and measured voltages match.

The last step of the calibration is the compensation of the phase shift between the excitation current and the coil voltages of S1 and S2. Ideally there should be a phase shift of 90${}^{\circ }$ since they are pure induced voltages caused by the primary magnetic field of P1. By using the power amplifier in between the ECFM and the LIA as the signal source, an additional phase shift with respect to the excitation current is introduced, which can be taken into account by setting a reference phase shift $\varphi_{\mathrm{ref}}$ that compensates for the influence of the power amplifier. Furthermore a slight phase shift is introduced by the eddy currents that are induced in the threaded rod at the centre of the sensor. For the sake of simplicity and for an easier comparison of the measurement results, a reference phase shift was chosen which sets the phase shift between induced voltages and excitation current to zero when the ECFM is surrounded by air. This last step is not necessary for the calibration, but it simplifies the handling and evaluation of the results. The adjusted simulation parameters that were obtained through calibration are listed in Table 1. Through this calibration, the mismatch for the induced voltages in S1 and S2 between model and real sensor was reduced below 0.05% for each of the three frequencies (500 Hz, 1 kHz, 2 kHz) that are used for the creation of the LuT. For P1, however, the mismatch lies in the range of several percent, caused by different factors like the internal resistance of the power amplifier as well as the inductance and resistance of the supply wires.

## 3. Results

### 3.1. Creation of the Look-Up-Table and Numerical Validation
of the Method

Now that the ECFM model is calibrated, it can be extended by a numerical model of the GaInSn loop, which adds the test section of the loop, filled with GaInSn and the sensor casing to the numerical model of the ECFM (see Figure 1d). In frame of the simulation model, a constant flow velocity of $v_{\mathrm{ECFM}}$ is assumed in the whole test section, but $v_{\exp}$ is still used as the reference in the LuT, since this velocity is indicated by the reference flow meter. The electrical conductivity of all stainless steel parts is set to 1.6 MS/m. Due to the simple setup of the test section, it can be represented reasonably well with this 2D axisymmetric model. The calculations are performed with COMSOL Multiphysics 6.1 assuming a constant, homogeneous flow velocity of the liquid metal, parallel to the sensor axis. In any practical application of this method, the velocity *v* and the electrical conductivity σ of the liquid metal are unknown and influence the induced voltages in the secondary coils of the ECFM, while the voltage of the primary coil is only influenced by σ, albeit by small amounts: for a change in σ of the liquid metal between 3 MS/m and 4 MS/m, the magnitude of $V_{P1}$ changes by 0.2%, whereas the magnitude of each secondary coil changes by nearly 10%. Using the magnitude $r_{P1}$ and phase shift $\varphi_{P1}$ may still be useful for determining *v* and σ of the liquid metal, as will be illustrated further below.

In total, the LuT contains the magnitudes and phase shifts of both secondary coils: $r_{S1}$, $r_{S2}$, $\varphi_{S1}$, $\varphi_{S2}$ as well as the magnitude and phase shift of P1 and the difference between the secondary coil magnitudes and phase shifts $\Delta r_{S}$ and $\Delta \varphi_{S}$. Traditionally, the voltage difference between the secondary coils is used for determining the flow velocity, since it has the highest sensitivity to changes in *v* and also cancels out common-mode interference. In this investigation 161 different values of σ between 1 MS/m and 5 MS/m, 81 different values of *v* between −0.5 m/s and 1.5 m/s at three different frequencies *f* of the excitation current (500 Hz, 1 kHz, 2 kHz) are used. This results in a total of 39,123 parameter combinations. The calculation of the resulting LuT takes only 45 min on a standard office computer (Intel i5 3 GHz (Intel, Santa Clara, CA, USA), 16 GB RAM). The LuT contains the eight above mentioned phase shifts and magnitudes for all 13,041 parameter combinations of *v* and σ times the three frequencies that are used. For an unknown parameter combination of *v* and σ—in this example, one particular combination from the LuT is chosen to be the “unknown” combination—the eight phase shifts and magnitudes are known for each of the three excitation frequencies. In a practical application of this method the voltages $V_{P1}$, $V_{S1}$ and $V_{S2}$ have to be measured. In order to determine the deviation between the voltages of the unknown parameter combination and all simulated parameter combinations, the mean squared error (MSE) is calculated for each of the eight phase shifts and voltages and three frequencies in the LuT. The MSE is calculated from the respective magnitudes and phase shifts of the unknown parameter combination.

In the following example the MSE for $V_{S1}$ is calculated. The MSE of the voltages with magnitudes and phase shifts for the unknown parameter combination (upc) $V_{S1,\mathrm{upc}}$ and all voltages of the LuT $V_{S1,\mathrm{LuT}}$ for a given number of different excitation frequencies $n_{f}$ has to be calculated. Since the induced voltages are strongly dependent on the excitation frequency, appropriate weighting factors *w* are introduced, which guarantee that each frequency has a similar impact on the total MSE. Otherwise high frequencies would dominate the MSE whereas low *f* would almost have no impact. The calculation of the MSE is done according to Equation (Equation 2) for $V_{P1}$, $V_{S1}$, $V_{S2}$ and $\Delta V_{S}$ for all k=1,...,13,041. Here, $V_{S1}$ is used as an example: (2)$${MSE}_{S1,k}^{2}=\frac{1}{n_{f}}\sum_{i=1}^{n_{f}}\frac{1}{w_{rS1,i}}{\left(r_{S1,\mathrm{upc},i}-r_{S1,\mathrm{LuT},i,k}\right)}^{2}+\frac{1}{w_{\varphi S1,i}}{\left(\varphi_{S1,\mathrm{upc},i}-\varphi_{S1,\mathrm{LuT},i,k}\right)}^{2}$$

Weighting the phase shifts is also useful since there is a noticeable difference in the phase shifts for primary and secondary coils, as well as for the voltage difference $\Delta V_{S}$. The most obvious choice for the weighting factors is the variance (var) of all magnitudes $r_{\mathrm{LuT}}$ or phase shifts $\varphi_{\mathrm{LuT}}$ in the LuT with the same coil and frequency. Alternatively, we also consider the squared median absolute deviation (mad) which is known to suppress extreme outliers. Just for comparison, we will also show a few results when using the standard deviation (std) or no weighting at all (*w* = 1). For each of the *i* frequencies, the weighting factors are calculated according to Equation (Equation 3) following the example of the weighting factor $w_{rS1}$ when using the variance for the magnitude of $V_{S1}$, where ${\bar{r}}_{S1}$ is the mean value of all $r_{S1}$ in the LuT with the same coil and frequency: (3)$$w_{rS1,i}=\frac{1}{N-1}\sum_{k=1}^{N=13,041}{\left(r_{S1,\mathrm{LuT},i,k}-{\bar{r}}_{S1,\mathrm{LuT},i}\right)}^{2}\;.$$

The weighting factors for the phase shifts are calculated in the same way by simply replacing the magnitudes by the according phase shifts. For the previous application of this method in [25], weighting by the variance yielded optimal results.

Each of the 13,041 parameter combinations of *v* and σ in the LuT now has four MSE (for $V_{P1}$, $V_{S1}$, $V_{S2}$ and $\Delta V_{S}$), that include the weighted values for each frequency. The total MSE called $MSE_{\Sigma }$ for a given parameter combination is obtained by summing up the four separate MSE according to
(4)$$MSE_{\Sigma ,k}=\sqrt{MSE_{S1,k}^{2}+MSE_{S2,k}^{2}+MSE_{P1,k}^{2}+MSE_{\Delta S,k}^{2}}\;.$$

Under ideal conditions and assuming that the numerical model has been properly calibrated, the parameter combination with the lowest total $MSE_{\Sigma }$ has the maximum likelihood to be the unknown parameter combination. By finding the minimum $MSE_{\Sigma }$ in the LuT, the values of *v* and σ can be inferred. When plotting all $MSE_{\Sigma }$ for a given unknown parameter combination (chosen from the LuT for now, not from the measurement) over *v* or σ it can be observed in Figure 3 that the MSE are converging to a certain minimum. In this case, the minimum of $MSE_{\Sigma }$ indicates the correct values of *v* and σ, since there are no measurement errors or other disturbances that have a negative impact on the results. Although this will certainly look different when using the actual measurements for the calculation of $MSE_{\Sigma }$, it can still be observed that this method is suitable to solve this problem and that the MSE are converging to the sought-after parameters. Measures for how well the MSE are converging are the angle α and the density of $MSE_{\Sigma }$ close to the minimum. Both small α and small density facilitate the parameter estimation of *v* and σ, since there are less parameter combinations with a similar $MSE_{\Sigma }$ that can be mistaken for the correct parameter combination in the presence of measurement errors.

In Figure 4 we illustrate how the $MSE_{\Sigma }$ for *v* (blue) and σ (violet) are changing when more and more voltages and frequencies are added, and how the weighting method is affecting the MSE distribution. Which components are used for the calculation is indicated by the text above each diagram. Greyed out components are not used in the calculation of $MSE_{\Sigma }$. The variance is used as the weighting method for the diagrams A–P and T in Figure 4, while diagrams Q, R and S use different weighting methods, which are indicated below each diagram. Starting from the left side in Figure 4, for only a single voltage at a single frequency with using either the magnitude or phase shift, there is no clear minimum of $MSE_{\Sigma }$ for *v* since several combinations of velocity and electrical conductivity can have the same magnitude or phase shift (Figure 4(A,B)). On the other hand, a certain range for σ can already be identified. By combining the (weighted) $MSE_{\Sigma }$ for *r* and φ, it becomes possible to identify *v* and σ when using a single coil voltage (C). When adding a second voltage (D–F) a clear minimum can be identified, with noticeable differences in $MSE_{\Sigma }$ for *r* and φ. For *r*, the MSE are converging faster to the minimum (D), which facilitates identification of the correct parameter combination. The density of MSE at the minimum is still comparably high, which means that there are multiple parameter combinations with a similar $MSE_{\Sigma }$, making it harder to identify the correct parameter combination. Adding a second frequency (K) or the voltage of the primary coil P1 (H) to $MSE_{\Sigma }$ does not have a significant influence on the results. However, adding the voltage difference of the secondary coils ΔS (G, J, L) improves the quality of the results significantly, since it depends sensitively on *v* and σ. The density of $MSE_{\Sigma }$ is strongly reduced at the minimum and is converging much better than in the previous cases. Adding a third frequency (P) or more voltages (N, O) does not yield additional benefits compared to cases G, J or L.

Regarding the weighting methods, using the variance (O) appears to be the optimal choice, exhibiting the lowest $MSE_{\Sigma }$ density and strongest convergence at the minimum. As expected, the worst results are achieved when not weighting at all (S), with the other methods (Q, R) in between where weighting by the squared median absolute deviation has a similar effect as using the variance. The reason why certain coil voltages have a stronger influence on the results than others can be seen in Table 2, where the sensitivity to changes in *v* and σ are displayed. While the magnitude and phase shift of P1 are barely influenced, ΔS, S1 and S2 are moderately sensitive to changes in σ. Moreover, ΔS is very sensitive to changes in the flow velocity compared to P1, S1 and S2, especially the phase shift. This is why ΔS has such a strong impact on the total MSE.

When using only ΔS for the calculation of $MSE_{\Sigma }$ (T), the identification of *v* and σ is possible, but more accurate results can be achieved when using multiple voltages and frequencies (O). The secondary coils S1 and S2 are more sensitive to changes in σ than ΔS but are considerably less sensitive to changes in *v*. Therefore, the $MSE_{\Sigma }$ distribution has in some cases different shapes, depending on whether *v* and σ is estimated (A–F). Adding P1 in the calculation of $MSE_{\Sigma }$ appears not to provide any improvement for the parameter estimation since its voltage is only slightly influenced by *v* and σ. This influence is of the same order of magnitude as the measurement accuracy and therefore measurement errors of $V_{P1}$ might significantly distort the results.

Once the LuT has been calculated, it takes less than 0.2 s to run the python script for calculation of the MSE and identification of the values for *v* and σ. However, in order to determine the unknown *v* and σ, the simulation model and the real setup have to match as well as possible. In the following section, the practical validation of this method is presented.

### 3.2. Experimental Validation

In the previous section it was shown that under ideal conditions it is straightforward to determine the unknown *v* and σ by using multiple voltages and frequencies. In reality, measurement errors, signal noise and the flow profile around the sensor and their influence on the measured voltages have to be considered and can have a negative impact on the results of the parameter estimation. The measurements at the GaInSn loop are conducted for six pre-adjusted flow velocities $v_{\exp}$ from 0 m/s to 1 m/s in steps of 0.2 m/s. The electrical conductivity of the liquid metal is assumed to be 3.3 MS/m as is described in [26] for pure GaInSn with a similar composition, although it might be contaminated with oxides or other impurities that have an unknown impact on σ. By using the LuT in combination with the measured voltages, the separate MSE for each magnitude, phase shift of $V_{P1}$, $V_{S1}$, $V_{S2}$ and $\Delta V_{S}$ and each excitation frequency can be calculated and weighted according to Equation (Equation 2). The sum of the separate weighted mean squared errors $MSE_{\Sigma }$ does not necessarily have to be calculated from all of the separate MSE. When leaving out, for example, all MSE of a certain frequency or by only using the MSE of the magnitudes or even leaving out a complete coil voltage when calculating $MSE_{\Sigma }$, the quality of the parameter estimation of *v* and σ can be influenced significantly (see Figure 4). How well a certain combination is performing is evaluated with the deviations Δv between pre-adjusted velocities $v_{\exp}$ and the velocity that is obtained by finding the minimum of the $MSE_{\Sigma }$ distribution $v_{\mathrm{LuT},\min}$, as well as the deviation Δσ between the electrical conductivity of the liquid metal σ and the conductivity that is obtained from the LuT $\sigma_{\mathrm{LuT},\min}$: (5)$$\Delta v=v_{\exp}-v_{\mathrm{LuT},\min}\;\;|\Delta v_{\mathrm{avg}}|=\frac{1}{6}\sum_{m=1}^{6}\left|v_{\exp,m}-v_{\mathrm{LuT},\min,m}\right|$$
(6)$$\Delta \sigma =\sigma_{\exp}-\sigma_{\mathrm{LuT},\min}\;\;|\Delta \sigma_{\mathrm{avg}}|=\frac{1}{6}\sum_{m=1}^{6}\left|\sigma_{\exp-\sigma_{\mathrm{LuT},\min,m}}\right|$$

Table 3 shows the absolute averages $|\Delta v_{\mathrm{avg}}|$ and $|\Delta \sigma_{\mathrm{avg}}|$ of all Δv and Δσ for each of the six $v_{\exp}$ for selected cases from Figure 4, when using the measured voltages instead of a set of voltages from the LuT itself and the two best weighting methods. Which of the components were used for the calculation of $MSE_{\Sigma }$ is indicated by the numbers 1 (=used), 0 (=not used). Here, the column headers indicate the kind of component. S1, S2, P1 and ΔS stand for the different voltages, *r* and φ stand for the magnitude and phase of the voltages. It is also shown which frequencies were used for the calculation as well as which weighting method (column header w) was used.

Although each of the investigated cases in Table 3 appears similar in Figure 4 when only using numerical data for the investigation of $MSE_{\Sigma }$, there are significant differences when using the actual measurement data for calculation of the MSE. By comparing Δv and Δσ for the different cases, it can be seen that some of them yield more accurate results than others, with cases N and T exhibiting the largest deviations in $v_{\exp}$ and $\sigma_{\exp}$, while cases G, J and M are among the most accurate ones.

Generally, weighting by variance or by squared median absolute deviation give comparable results, with no clear advantage offered by one method over the other. Case T, which represents the bare minimum case where only one voltage and one frequency is used for the estimation of *v* and σ, gives relatively inaccurate results. Regarding the velocity estimation, case J is more accurate than case T by a factor of 4, and regarding the electrical conductivity accuracy is better by a factor of 7. Notice however, that the electrical conductivity of the liquid metal is not exactly known. From the overall results it can be deduced that at least two voltages, both the magnitude and the phase shift, and one frequency have to be used for the calculation of $MSE_{\Sigma }$ for obtaining optimal results. Although ΔS has the highest impact for increasing the accuracy of the parameter estimation, it is not sufficient to only use ΔS, rather the individual voltages of S1 and S2 should be used in addition. By contrast, adding the voltage of the primary coil P1 does not give any benefits and would also require more effort since an additional voltage has to be measured. Using one frequency from the range where the ECFM is most sensitive seems to be sufficient, while introducing a second or third frequency appears to offer no advantages for this application and can even be detrimental in some cases (because additional measurement errors are introduced). Using either *r* or φ is possible but since the two quantities are always measured simultaneously when using a LIA, there are no disadvantages when using both for the calculation of $MSE_{\Sigma }$.

In Figure 5 the velocity estimations of case G and case J from Table 3 are compared when using the squared median absolute deviation weighting method. Case G converges faster to the minimum, which is also much lower than for case J. Ultimately, the value of the minimum is unimportant, as long as it can be clearly identified. Due to measurement errors and the differences between the real setup and the simplified simulation model, $MSE_{\Sigma }$ does not reach zero. When comparing Δv it can be seen that the deviations of case G are distributed more uniformly for different $v_{\exp}$ than for case J where the maximum deviation is found at zero velocity and is progressively lower for higher velocities. Regarding the estimation of the electrical conductivity, the $MSE_{\Sigma }$ in Figure 6 are similar but converge more slowly than for the flow velocity. This is in good agreement with the purely numerical results in Figure 4. While case J yields the lowest Δv overall, the average deviation of σ is smaller for case G (but note again that the “real” value of σ is not well known).

In order to assess the influence of the individual $MSE_{\Sigma }$ components, the evolution of $MSE_{\Sigma }$ is traced in Figure 7 by using the measurement data, similar to the more extensive investigation in Figure 4. It can be seen that for case A the flow velocity cannot be identified at all and the estimated conductivity is far too low, even outside of the displayed range of σ. When adding the measurements of the phase shift to $MSE_{\Sigma }$ a velocity which is far greater than its actual value can be identified, while the σ estimation is now much closer to the actual value. By adding the voltage of the other secondary coil in case F, the velocity estimation is much improved, while the estimated σ is slightly worse. Adding ΔS in case J only has a small influence on the accuracy of the parameter estimation in this case but significantly reduces the $MSE_{\Sigma }$ density at the minimum and also has the effect that $MSE_{\Sigma }$ converges faster. When finally adding a second frequency in case M, the velocity estimation as well as the estimation of the electrical conductivity become more accurate.

### 3.3. Measurement Errors

#### 3.3.1. Inductance

The inductances of the sensor coils are measured with a Hameg HM8118 LCR-Bridge. According to the data-sheet, the measurement accuracy depends on the impedance of the coils, and therefore the inductance of P1 has an accuracy of ±0.13%, and for S1, S2 an accuracy of ±0.103%. For all coils, the resulting measurement error corresponds to a change in less than one turn for the inductance of the coils and is therefore sufficiently accurate for the calibration of the numerical model.

#### 3.3.2. Voltage

For the voltage measurements, a Signal Recovery 7265 DSP Lock-in Amplifier (Signal Recovery, Irvine, CA, USA) is used. According to the data sheet, it has an accuracy of ±0.3% for the measured voltages. Each coil voltage (magnitude and phase shift) is measured for one minute with one data point per second for a constant velocity $v_{\exp}$ and frequency. The final voltage, which is used for the calculation of the MSE, is calculated from the mean value of all data points. The measured voltages are almost constant, as can also be seen on the standard deviation of the data series: for every measured voltage it lies below 0.02%, and therefore the random error caused by noise or other electromagnetic disturbances is negligible. In a practical application of this measurement technique, it is sufficient to measure only one data point.

#### 3.3.3. Flow Rate

For measurements of the reference flow rate $v_{\exp}$ of the loop, an ABB Copa-XL Electromagnetic Flowmeter (ABB, Zurich Switzerland) is used. According to the data-sheet, the accuracy for the velocity measurements is ±0.5% of the flow velocity when $v_{\exp}$ < 0.7 m/s, and ±0.0035 m/s for $v_{\exp}$ > 0.7 m/s. Like for the coil voltages, $v_{\exp}$ is calculated from the mean value of the flow velocities over one minute, with one measurement made every second. For every measured $v_{\exp}$, the standard deviation lies below 0.3%. These values apply for the mean velocity in the pipe. The actual velocity profile of the flow cannot be measured since both the reference flow meter and the ECFM are inductive sensors that are only able to measure the mean flow rate or flow velocity in a certain volume. In addition to the measurement accuracy, the quality of the calibration of the flowmeter has a significant influence on the total measurement error. Since the flow meter was calibrated many years ago when the GaInSn loop was constructed, there might be an additional systematic error of unknown dimensions for all the velocity measurements.

#### 3.3.4. Electrical Conductivity of the Liquid Metal

The electrical conductivity of the eutectic liquid metal alloy GaInSn in the loop is not exactly known, due to the fact that it might contain oxides or other impurities. Depending on the volume concentration of impurities and their electrical conductivities, the total electrical conductivity of the liquid metal might vary according to Equation (Equation 1) in [28]. Assuming that the liquid metal has the properties of pure GaInSn as described in [26], it has an electrical conductivity of 3.3 ± 0.3 MS/m at room temperature. It is assumed that there is only a small amount of impurities contained in the liquid metal and their influence on the total electrical conductivity does not exceed the relatively large measurement uncertainty of 0.3 MS/m.

## 4. Conclusions

By combining numerical simulations and experimental results, we have shown that the flow velocity and the electrical conductivity of a liquid metal can be obtained simultaneously when using an ECFM in combination with a so called Look-Up-Table. This method may not be as accurate as other measurement techniques like a properly calibrated ECFM or an ultrasound doppler velocimetry sensor, but it can be used as an alternative for the time consuming temperature calibration of the ECFM, while also eliminating the need for a temperature measurement of the liquid metal. In order to achieve accurate results, a model for numerical simulations has to be created that matches the real setup as precisely as possible. Additionally, the simulation model of the ECFM has to be calibrated by performing measurements at the real sensor, prior to inserting it into the test section or experimental facility. Once the ECFM is calibrated and the Look-Up-Table has been calculated, the flow velocity and electrical conductivity of the liquid metal can be estimated in less than a second. This time span can be decreased further by reducing the resolution and range of the Look-Up-Table parameters. When the same sensor is put into a different environment, no further calibration of the ECFM is necessary. Only the relevant part of the facility has to be translated into a numerical model, since the calibration of the sensor in air is sufficient. At our liquid metal facility, we were able to achieve an average deviation between real and estimated velocity of 0.04 m/s in a range between 0 m/s and 1 m/s and for the electrical conductivity an average deviation of 4 ×$10^{4}$ S/m. Using multiple excitation frequencies for the sensor is not necessary. Similar results are obtained, regardless of how many excitation frequencies are used. Weighting the mean squared errors is also important, and in this experiment using the variance or squared median absolute deviation provided the best results. Accurate results were achieved when using only one excitation frequency in the sensitive range of the ECFM. Yet, for other applications multiple frequencies might yield better results, depending on the setup and on whether there are components with different electrical conductivities within the range of the sensor. This measurement technique can be applied to similar problems and/or sensors and it is also possible to infer related parameters like the temperature of the liquid metal, for example. As long as the electrical conductivity of the liquid is in the order of magnitude of 0.1 MS/m or more, this method can easily be applied to other liquid metals and alloys.

In summary, the main achievements of this paper are the following:First simultaneous measurement of flow velocity and electrical conductivity of a liquid metal using an ECFM.Simplified calibration of the sensor compared to conventional methods.Parameter estimation of velocity and conductivity takes less than a second, using only one excitation frequency.This method can be applied to many different liquid metals and alloys, different parameters and sensors.

## Figures and Tables

**Figure 1 sensors-23-09018-f001:**
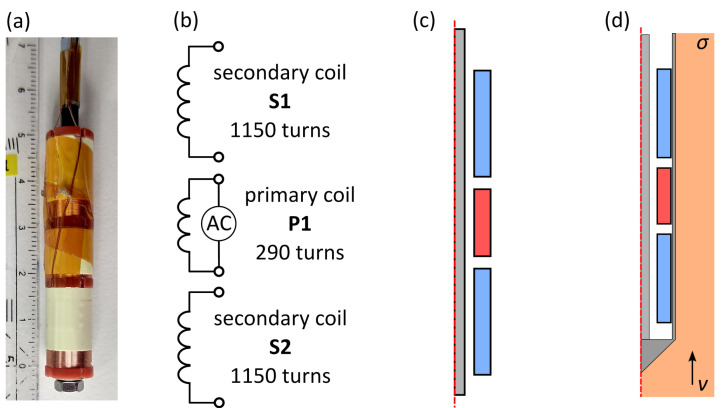
(**a**) Photograph, (**b**) coil overview with specifications and (**c**,**d**) simulation models of the ECFM.

**Figure 2 sensors-23-09018-f002:**
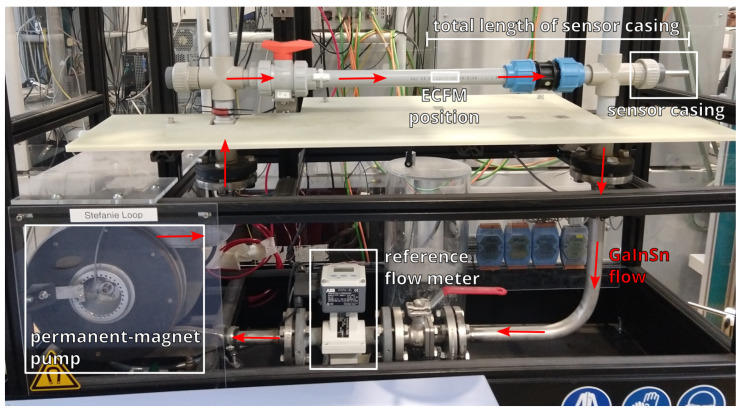
Photograph of the GaInSn loop and measurement setup. The red arrows indicate the flow direction of the liquid metal.

**Figure 3 sensors-23-09018-f003:**
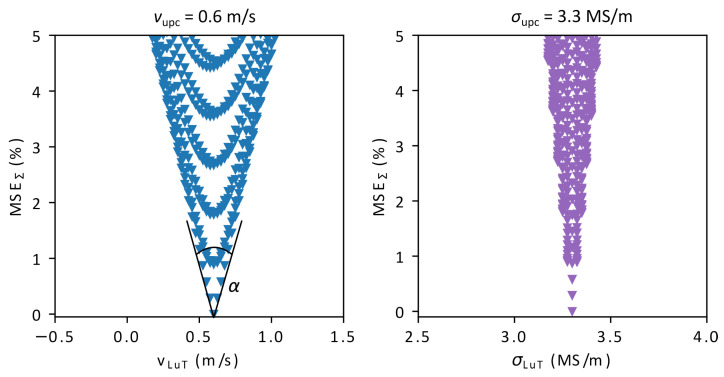
Distribution of the lowest 5% of $MSE_{\Sigma }$ for *v* and σ under ideal conditions, when taking one parameter combination of the LuT as the unknown parameter combination (*v* = 0.6 m/s, σ = 3.3 MS/m).

**Figure 4 sensors-23-09018-f004:**
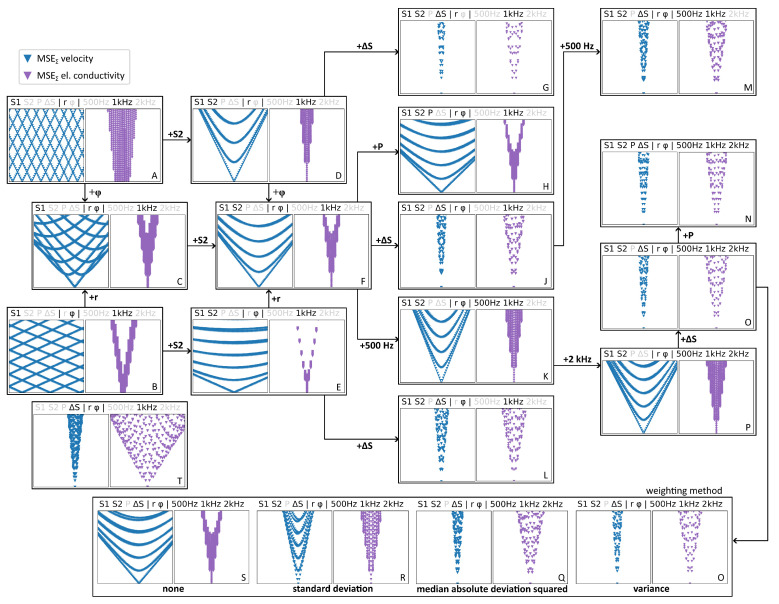
Evolution of the lowest 5% of $MSE_{\Sigma }$ for *v* and σ (similar to Figure 3) when adding more coil voltages, frequencies or changing the weighting method. Blue triangles represent $MSE_{\Sigma }$ of the flow velocity, violet triangles represent $MSE_{\Sigma }$ of the electrical conductivity.

**Figure 5 sensors-23-09018-f005:**
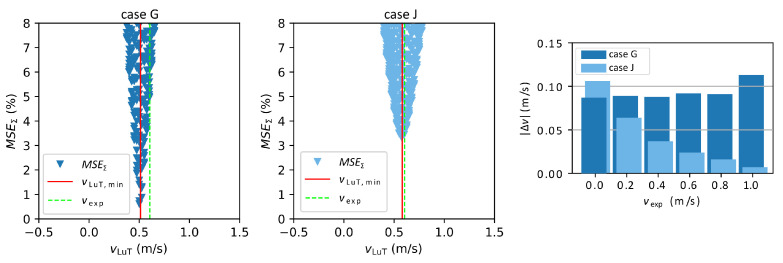
Distribution of $MSE_{\Sigma }$ of the flow velocity for $v_{\exp}$ = 0.605 m/s for cases G and J of Table 3. The red line indicates the minimum of the MSE distribution while the dashed green line shows the pre-adjusted flow velocity. The diagram on the right shows a comparison of Δv for all $v_{\exp}$.

**Figure 6 sensors-23-09018-f006:**
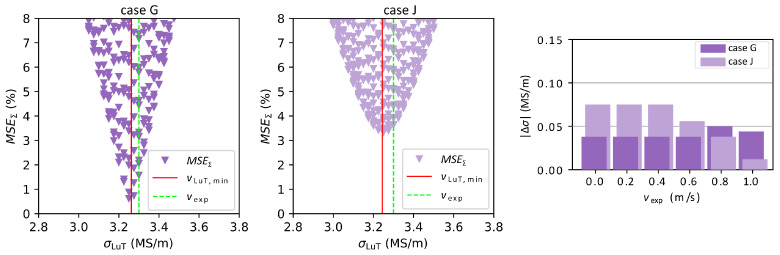
Distribution of $MSE_{\Sigma }$ of the electrical conductivity for $v_{\exp}$ = 0.605 m/s for cases G and J of Table 3. The red line indicates the minimum of the MSE distribution while the dashed green line shows the electrical conductivity of the liquid metal. The diagram on the right shows a comparison of Δσ for all $v_{\exp}$.

**Figure 7 sensors-23-09018-f007:**
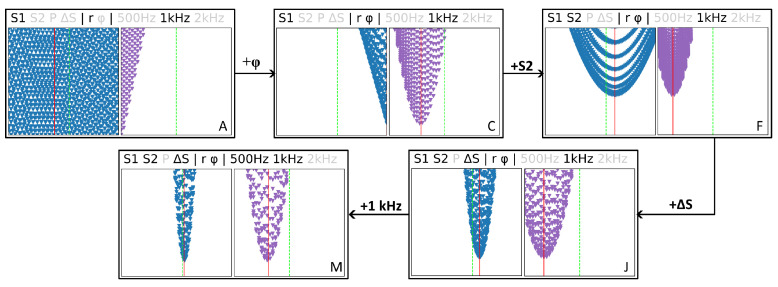
Evolution of $MSE_{\Sigma }$ from case A to case M when using the measurement data. The red line indicates the minimum of the MSE distribution while the dashed green line shows the pre-adjusted flow velocity or electrical conductivity. Blue triangles represent $MSE_{\Sigma }$ of the flow velocity, violet triangles represent $MSE_{\Sigma }$ of the electrical conductivity.

**Table 1 sensors-23-09018-t001:** Results of the calibration of the three sensor coils.

Coil	*L* (mH)	$n_{\mathrm{spec}}$	$n_{\mathrm{cal}}$	$\Delta d_{P1}$ (μm)	$\varphi_{\mathrm{ref}}$ at 0.5, 1, 2 kHz
P1	0.312	290	283	-	-
S1	3.37	1150	1100	69	83.6${}^{\circ }$, 77.72${}^{\circ }$, 65.86${}^{\circ }$
S2	3.41	1150	1108	−4	

**Table 2 sensors-23-09018-t002:** Relative changes in magnitude and phase shift for the voltages of S1, S2, ΔS and P1 for a constant velocity (0 m/s) and a change in electrical conductivity of 1 MS/m (from 3 to 4 MS/m), as well as for a constant electrical conductivity (3 MS/m) and a change in velocity of 1 m/s (from 0 to 1 m/s) at 1 kHz.

Conditions	S1, S2		ΔS		P1	
	**r**	φ	**r**	φ	**r**	φ
*v* = 0 m/s	8.1%	20.6%	5.2%	30.8%	0.2%	1.9%
σ = 3 → 4 MS/m						
*v* = 0 → 1 m/s	0.8%	0.6%	62.1%	111.5%	0.0%	0.0%
σ = 3 MS/m						

**Table 3 sensors-23-09018-t003:** Results of Δv and Δσ for the cases G, J, L, M, N, O and T of Figure 4 when weighting by variance (var) and squared median absolute deviation (mad). The color highlights the used components.

Case	$|\Delta v_{\mathrm{avg}}|$ (cm/s)	$|\Delta \sigma_{\mathrm{avg}}|$ (MS/m)	S1	S2	P1	ΔS	*r*	φ	500 Hz	1 kHz	2 kHz	w
G	9.3	0.042	1	1	0	1	1	0	0	1	0	var
	9.3	0.041										mad
J	5.6	0.064	1	1	0	1	1	1	0	1	0	var
	4.2	0.055										mad
L	11.1	0.130	1	1	0	1	0	1	0	1	0	var
	10.9	0.127										mad
M	4.3	0.196	1	1	0	1	1	1	1	1	0	var
	6.0	0.185										mad
N	20.4	1.173	1	1	1	1	1	1	1	1	1	var
	26.5	1.133										mad
O	7.0	0.410	1	1	0	1	1	1	1	1	1	var
	6.6	0.490										mad
T	15.8	0.497	0	0	0	1	1	1	0	1	0	var
	15.9	0.487										mad

## Data Availability

The data that support the findings of this study are available from the corresponding author, F.S., upon reasonable request.

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
