# Peer review of "Simultaneous Measurement of Flow Velocity and Electrical Conductivity of a Liquid Metal Using an Eddy Current Flow Meter in Combination with a Look-Up-Table Method"

_sensors, 2023, doi:10.3390/s23229018_

Round 1

Reviewer 1 Report

Comments and Suggestions for Authors

In the manuscript “Simultaneous measurement of flow velocity and electrical conductivity of a liquid metal using an Eddy Current Flow Meter in combination with a Look-Up-Table method” presented a method that allows the simultaneous measurement of electrical conductivity and flow velocity, without prior knowledge of the exact conductivity or temperature of the liquid metal, by using a so called Look-Up-Table method. The viability of this measurement technique is substantiated by numerical simulations and validated by experiments that were performed in the liquid metal alloy GaInSn at room temperature.

There are minor comments about the work:

1. The work very superficially discusses the consequences of contamination of the GaInSn alloy and does not discuss how the measured parameters will be affected by impurities.

2. The conclusions do not include the possibility of expanding this approach to other alloys, which I would like to see.

Comments on the Quality of English Language

Minor editing of English language required.

Reviewer 2 Report

Comments and Suggestions for Authors

In this manuscript, the authors used the look-up-table method to derive the velocity and electrical conductivity of a metal simultaneously. The authors conducted experiments and also numerical methods to derive the table, validated the results, and analyzed the measurement error. The advantage of the method is its efficiency and simultaneous derivation, while the disadvantage is that it may not be as accurate as direct measurement. The method described in the manuscript has some applications if the requirement of accuracy is not strict. There are some questions that the authors need to answer before the manuscript can be accepted.

1, Is the look-up-table essentially describing the relationship between parameters v, sigma, f with output voltages? If so, is it possible to conduct fitting between these variables such that the velocity and sigma can be easier to derive?

2, Line 69, In line 54, it is mentioned that there are three excitation frequencies used, then here in line 69 the actual frequencies should be listed. The range 500 Hz to 2 kHz is not clear enough.

3, Can the authors provide an image about the liquid metal loop to make it easy to understand about the experimental setup? 

4, Line 102, how is the accuracy of the 2D model comparing to the 3D model? What’s the time cost for simulating a 2D case and a 3D case?

5, Can the authors show an example of the look-up-table? I did not find it.

6, Line 386, what is the comparison of accuracy comparing to other measurement techniques?

7, Although during the measuring period the method is efficient to derive the velocity and electrical conductivity, but the creation of the look-up-table may be a big project. How does the authors justify the overall efficiency of the method?

Reviewer 3 Report

Comments and Suggestions for Authors

Dear Authors,

Many thanks for your submitted paper. It was interesting and had a good structure. However, it suffers from some aspects which are listed as below:

1- The abstract must be modified and some quantitative values must be provided. You need to provide readers some numbers or percentage and motivate them that your method is better! 

2- What is the novelty of your work? It is the only solution or substitution? Nobody has mentioned it before? 

3- You need to provide a nomenclature table for this work as some of the terms are not defined properly.

4- Please mention your conclusions in bullet format to help readers to see your major concluding remarks! 

5- The English language of the paper must be refined by a native to remove the typos. Please remove the direct tense and mostly use passive forma in Scientific writing.

6- More updated references must be added to increase the strength of the introduction part.

Comments on the Quality of English Language

The English language of the paper must be refined by a native to remove the typos. Please remove the direct tense and mostly use passive forma in Scientific writing.
